# Imitating What Works: Simulation-Filtered Modular Policy Learning from Human Videos

**Albert J. Zhai**[1]  **Kuo-Hao Zeng**[2]  **Jiasen Lu**[2]  **Ali Farhadi**[2,3]  **Shenlong Wang**[1]  **Wei-Chiu Ma**[4]

[1] *University of Illinois Urbana-Champaign*
[2] *Allen Institute for AI*
[3] *University of Washington*
[4] *Cornell University*

**Reviewed on OpenReview:** *https://openreview.net/forum?id=ZEmv4DhaGL*

## Abstract

The ability to learn manipulation skills by watching videos of humans has the potential to unlock a new source of highly scalable data for robot learning. Here, we tackle prehensile manipulation, in which tasks involve grasping an object before performing various post-grasp motions. Human videos offer strong signals for learning the post-grasp motions, but they are less useful for learning the prerequisite grasping behaviors, especially for robots without human-like hands. A promising way forward is to use a modular policy design, leveraging a dedicated grasp generator to produce stable grasps. However, arbitrary stable grasps are often not task-compatible, hindering the robot's ability to perform the desired downstream motion. To address this challenge, we present *Perceive-Simulate-Imitate* (PSI), a framework for training a modular manipulation policy using human video motion data processed by paired grasp-trajectory filtering in simulation. This simulation step extends the trajectory data with grasp suitability labels, which allows for supervised learning of task-oriented grasping capabilities. We show through real-world experiments that our framework can be used to learn precise manipulation skills efficiently without any robot data, resulting in significantly more robust performance than using a grasp generator naively.

## 1 Introduction

Learning from large-scale data has proven to be a successful strategy for building highly general and effective models for many tasks in computer vision and natural language processing (Brown et al., 2020; Radford et al., 2021; Kirillov et al., 2023; Blattmann et al., 2023; Yang et al., 2024; Wang et al., 2024). A number of works have provided evidence to suggest that the same holds true for robotics, with unprecedented performance levels being achieved through larger and larger datasets (Brohan et al., 2022; Walke et al., 2023; O'Neill et al., 2024; Ghosh et al., 2024; Khazatsky et al., 2024; Kim et al., 2024; Black et al., 2024). However, the scale of such datasets are ultimately still limited due to the high cost and difficulty of obtaining high-quality real-world robot action data.

Recently, there has been growing interest in a potential workaround for scalable data acquisition: learning by watching humans. Studies in psychology and cognition have shown that human babies are able to watch others perform actions and then imitate them (Meltzoff & Moore, 1977; Meltzoff, 1988; Tomasello et al., 1993). Inspired by this, many works aim to allow robots to do the same. Acquiring this ability has the potential to transform the field of robot learning, because not only is it fairly inexpensive to collect new videos of humans, there is also an abundance of already existing human video data on the internet.

In this work, we study the problem of learning robot manipulation policies purely from human videos. We focus on prehensile manipulation tasks – that is, tasks that involve grasping an object. Such tasks can be broken down into two subtasks: 1) finding an appropriate grasp, and then 2) performing an appropriate

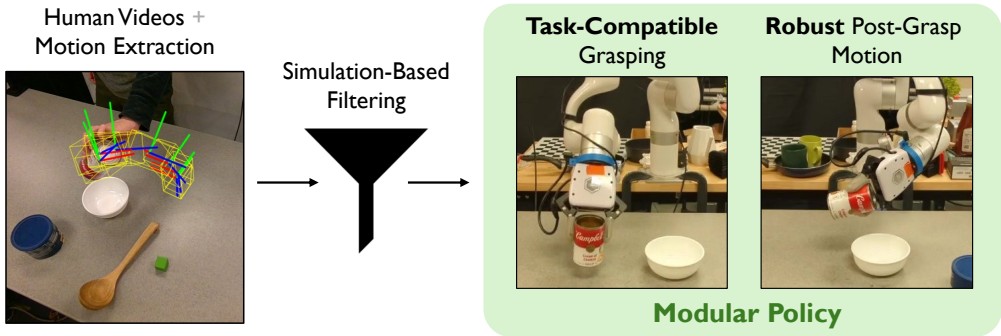

Figure 1: **Modular prehensile imitation learning.** Human videos are well-suited for learning post-grasp motions but are not suitable for learning grasping for non-anthropomorphic end-effectors. Separating these subtasks via a modular policy design allows for dedicated post-grasp learning. However, existing methods under this paradigm fail to acquire task-compatible grasping skills and are not robust to poor-quality motion data. We propose a simple but effective solution to these issues using simulation-based filtering and a learned grasp scoring model.

post-grasp motion to complete the task. For robots with anthropomorphic hands, both behaviors can potentially be learned in a unified manner by retargeting human hand poses (Qin et al., 2022; Chen et al., 2022; Shaw et al., 2023; Qiu et al., 2025). For robots with non-anthropomorphic end-effectors, such as standard parallel-jaw grippers, retargeting is much less effective. Existing works that attempt to learn both subtasks jointly for non-anthropomorphic grippers (Wen et al., 2023; Bharadhwaj et al., 2024b; Xu et al., 2024; Bharadhwaj et al., 2024a) have generally resorted to (costly) robot demonstrations to overcome the embodiment gap in grasping. Taking a *modular* approach, in which the subtasks are separated (Ko et al., 2024; Yuan et al., 2024; Shi et al., 2025), allows one to offload grasping to existing grasp generators and focus on modeling the post-grasp motions in human videos. This alleviates the reliance on costly robot demonstrations. However, current modular methods ignore the dependencies between the two subtasks and thus fail to ensure their grasping is task-compatible (Fig. 2). We propose a modular framework that uses simulation to overcome this hurdle, allowing for robust, sample-efficient learning using only human videos, as illustrated in Fig. 1.

We first dive deeper into the grasping subtask of the problem. Learning grasping skills from human videos for arbitrary robot embodiments is challenging due to the embodiment gap. Unless we assume our robot has a human-like hand, it is difficult to derive grasping supervision from human grasping behavior. For this reason, many previous works have abstracted away the grasping aspect of the problem, assuming that it can be handled by external grasp generators (Ko et al., 2024; Yuan et al., 2024; Shi et al., 2025; Liang et al., 2024). We call this the *modular* policy approach, and we agree that it is promising – modern grasp generators indeed exhibit the ability to find stable grasps on a wide variety of objects. However, in order for a grasp to be successful, it not only needs to be stable, but *task-compatible* as well. As illustrated in Fig. 2, many grasps are stable but prohibit the downstream motion to be performed. Achieving robust prehensile manipulation requires selecting grasps that are both stable and task-compatible.

Before we discuss our solution for achieving task-compatible grasping, let's first discuss the second subtask: learning *post-grasp* motions from human videos. This has been explored by a variety of recent works (Bahl et al., 2023; Bharadhwaj et al., 2024a; Wen et al., 2023; Liang et al., 2024; Ko et al., 2024; Bharadhwaj et al., 2024b; Xu et al., 2024; Yuan et al., 2024; Ren et al., 2025). A key insight here is that the success in manipulation largely depends on the *motion of objects* in the scene. Inducing the same object movement as in a successful demonstration should result in success, regardless of whether the agent is a human or a robot. Thus, if we can extract a representation of object motion from human videos, we can use it as a learning target for a policy model, and then convert the predictions to robot actions at test time. A number of works have used a flow-based representation (Ko et al., 2024; Bharadhwaj et al., 2024b; Xu et al., 2024; Yuan et al., 2024), which can be converted to $SE(3)$ end-effector actions by solving an optimization problem. In this work, we follow the more direct approach of using the 6-DoF object pose, which has also been shown to be promising (Lum et al., 2025; Hsu et al., 2025). We experiment with both model-based and model-free

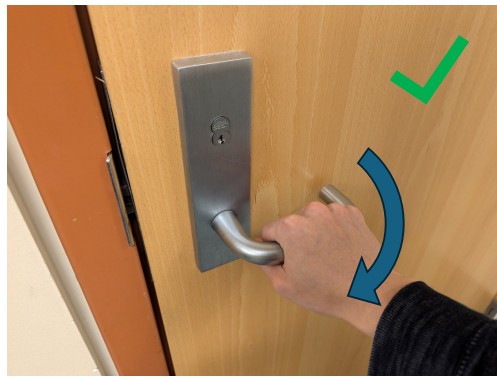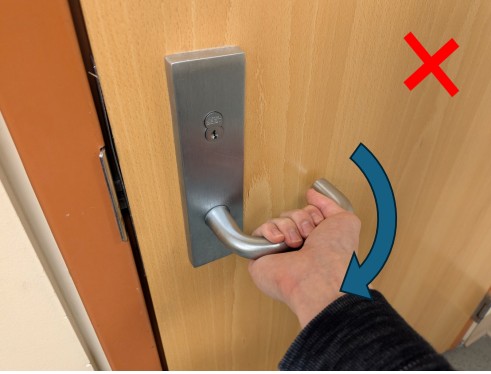

Figure 2: **Task-compatibility for grasps.** Even though a grasp may be stable, it may not be compatible with the downstream task. With a firm right hand underhand grip on the door handle (right), it becomes very difficult to turn the handle clockwise. Task-agnostic grasp generators fall short in solving this problem, highlighting the need for task-oriented grasping.

methods for extracting 6D pose trajectories. These can be cleanly converted via rigid transformation to end-effector actions given a grasp pose at test time.

Unfortunately, many of these extracted trajectories are actually unsuitable for robot learning. First of all, 6D pose tracking methods may fail and give erroneous trajectories that harm policy performance. Furthermore, even when the tracking is accurate, the motion achieved by the human might not be feasible *for the robot*, and thus should not be imitated. Therefore, in order to ensure that only the suitable motion data is used for imitation learning, it would be highly desirable to have a way to *filter out* trajectories.

We now introduce our solution for both trajectory filtering and task-compatible grasping: simulation. Given a set of 6D trajectories extracted from human videos, we pair each trajectory with multiple candidate grasps and execute the grasp-trajectory pairs on a simulated robot to determine which actions are suitable for that robot. Not only does this tell us which trajectories are completely infeasible, it also gives us per-trajectory grasp success labels that provide grasp supervision for our policy model. Specifically, we use simple behavior cloning to train a model that takes in (real) images and outputs both grasp suitability and post-grasp trajectories. At test time, we can leverage existing grasp generators for grasp *stability*, while employing our learned grasp-scoring model to ensure *task-compatibility*. This capability has not been achieved in previous works on visual imitation from humans for non-anthropomorphic end-effectors, which either rely on robot data to learn grasping or suffer from task-incompatible grasping.

Our overall framework, named Perceive-Simulate-Imitate (PSI), can teach robots manipulation tasks directly from human videos without any specialized equipment or tools. We evaluate its effectiveness relative to baselines across four manipulation tasks in the real world, considering both model-free and model-based pose tracking pipelines. Importantly, we show that without our simulation-based filtering, noisy trajectory data and task-incompatible grasp selection lead to high failure rates. Besides training new policies from scratch, PSI can also be used to pretrain policies directly on existing hand-object interaction datasets. We show that pretraining with PSI on HOI4D (Liu et al., 2022b) enables more sample-efficient learning downstream. We also perform experiments comparing performance across different robot embodiments and motion representations.

## 2 Related Work

**Learning manipulation skills from human videos** Imitation learning is a promising paradigm for learning manipulation policies, but robot data collection schemes such as teleoperation or kinesthetic teaching require complex and costly embodiment-specific setups. Human (hand-object interaction) videos are much easier to collect, so it is highly desirable to use them to augment or replace robot data for imitation learning. For dexterous robots with human-like hands, robot actions can be obtained by hand-pose estimation and retargeting (Sivakumar et al., 2022; Qin et al., 2022; Shaw et al., 2023; Singh et al., 2024; Fu et al., 2024; Li et al., 2024; Qiu et al., 2025). However, for robots with different embodiments, such as arms with two-finger

grippers, there is no straightforward conversion from human poses to robot poses. Some works have tried to retarget specific parts of human hands to the robot (Bharadhwaj et al., 2023; Papagiannis et al., 2024; Ren et al., 2025; Shi et al., 2025; Liu et al., 2025; Lepert et al., 2025b), but ultimately this is highly limited in the interactions it can handle. Others have tried to use image translation to acquire robot videos that can be converted to reward functions for reinforcement learning (Liu et al., 2018; Smith et al., 2020; Xiong et al., 2021), but this also suffers from poor generalization. Faced with the embodiment gap, many works have turned to learning different forms of manipulation information that are *related to* but not directly equivalent to robot actions. This includes visual features (Nair et al., 2023; Majumdar et al., 2023; Radosavovic et al., 2023), latent plans (Wang et al., 2023; Ye et al., 2025), local affordances (Bahl et al., 2023; Kuang et al., 2024; Bahl et al., 2022), future masks (Bharadhwaj et al., 2024a), and future motion (Bharadhwaj et al., 2024b; Wen et al., 2023; Xu et al., 2024; Hsu et al., 2025; Kareer et al., 2025). These methods use the learned information to guide policy learning, but they still rely on robot data for action supervision. Im2Flow2Act (Xu et al., 2024) allows the robot data to be provided in simulation, but it still requires access to an expert action generator. Our framework aims to learn manipulation skills without *any* reliance on robot demonstrations.

A few works have successfully trained manipulation policies using only human (cross-embodiment) videos and no robot data. General-Flow (Yuan et al., 2024) predicts 3D flow and then obtains post-grasp $SE(3)$ trajectories by performing least-squares alignment. AVDC (Ko et al., 2024) predicts video and 2D flow, and then solves for $SE(3)$ via PnP. Dreamitate (Liang et al., 2024) predicts video and then extracts $SE(3)$ via MegaPose (Labbé et al., 2022). ZeroMimic (Shi et al., 2025) directly predicts human wrist trajectories. All of these methods assume a modular policy in which grasping is offloaded to an external grasping model. However, by abstracting away the problem of grasping from the downstream motion, these methods fail to perform *task-oriented* grasping. Our framework aims to fix this issue by evaluating grasp-trajectory pairs in simulation and building a modular policy that incorporates a grasp scoring function.

**Simulation-based filtering for robot learning**  Simulation has been used to filter robot learning data before, but not for the purpose of learning task-compatibility from cross-embodiment video data. Physical simulators and analytical models have been used extensively to determine grasp stability (Kappler et al., 2015; Mahler et al., 2019). DexTransfer (Chen et al., 2022) uses a simulator to refine retargeting data for a dextrous hand. Inspired by the notion of selective imitation in psychology studies (Brody & Stoneman, 1981), SILO (Lee et al., 2020) proposes a reinforcement learning framework to learn which timesteps of a demonstration should be imitated. In this work, we propose a simulation-based filtering scheme in order to learn a grasp scoring model that can overcome the task-compatibility issue encountered by modular prehensile manipulation policies.

**Task-oriented grasping**  Task-oriented grasping refers to the ability to choose grasps that are suitable for a downstream manipulation task, as opposed to task-agnostic grasping, which is only concerned with grasp robustness (stability). Classical works studied task-oriented grasping based on task wrench spaces (Li & Sastry, 1988; Haschke et al., 2005). More recently, various methods have been proposed to learn task-compatibility information from labeled data (Murali et al., 2021; Antanas et al., 2019; Detry et al., 2017; Tang et al., 2023; 2025) or interaction (Fang et al., 2020; Qin et al., 2020; Agarwal et al., 2023). We propose a simple way to learn task-oriented grasping capabilities in a cross-embodiment imitation framework by evaluating grasp-trajectory pairs in simulation.

## 3 Method

### 3.1 Overview

Our goal is to enable robots to learn prehensile manipulation skills solely from RGB-D videos of humans manipulating objects. Our proposed solution, illustrated in Fig. 3, is a framework consisting of three steps: Perceive, Simulate, and Imitate. In the Perceive step, we use 3D vision techniques to track the 6D pose of active objects in videos (Sec. 3.2). The 6D pose trajectories capture the object motions needed to complete a task, and they can be directly translated to end-effector actions on a robot. However, many of the trajectories may be erroneous or infeasible for robot execution. In addition, the object motions do not provide supervision

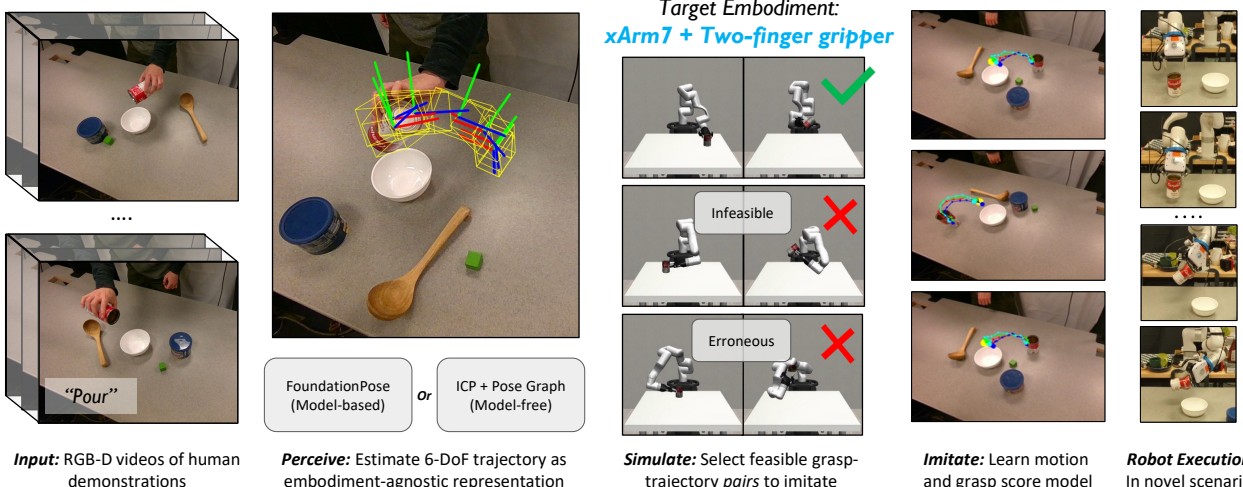

Figure 3: **Overview of our framework.** PSI is a three-step framework for visual imitation learning using only RGB-D videos of human demonstrations. In the Perceive step, we use 3D vision techniques to track the 6D pose of active objects in videos (Sec. 3.2). The 6D pose trajectories can be directly translated to end-effector actions on a robot. In the Simulate step, we leverage simulation to refine and enhance the pose trajectory data with grasp suitability labels (Sec. 3.3). In the Imitate step, we train an open-loop visuomotor policy on the data via behavior cloning (Sec. 3.4). The policy can be combined with any existing grasp generator to perform task-oriented grasping and manipulation on a real robot (Sec. 3.5).

regarding how to grasp the objects. This motivates the next step: Simulate, in which we leverage simulation to filter the pose trajectory data to be suitable for robot learning, as well as extract supervisory grasp labels (Sec. 3.3). Finally, in the Imitate step, we train a model that outputs both post-grasp motion and grasp scores (Sec. 3.4). The model can be combined with any existing grasp generator in a modular manner for real-world execution (Sec. 3.5).

## 3.2 Object 6-DoF Pose as Motion Representation

As mentioned in Sec. 1, the success of manipulation tasks largely depends on the *motion of objects* in the scene. Inducing the same object movement as in a successful episode will likely result in success again (given the same starting state), regardless of the specific end-effector or embodiment. We thus choose to abstract away body-specific information and instead represent human demonstrations through object motions.

**6-DoF Pose *vs.* Flow** There are numerous ways to describe the motion of objects. One predominant representation that has received widespread attention is *flow* (Ko et al., 2024; Bharadhwaj et al., 2024b; Xu et al., 2024; Yuan et al., 2024). Flow is a highly general motion representation, but in order to train manipulation policies, we ultimately need to convert the flows into an action space such as $SE(3)$ end-effector transformations. 6-DoF poses can be directly converted into end-effector actions via a rigid transform, eliminating the need for an additional depth-based conversion step (Ko et al., 2024; Bharadhwaj et al., 2024b; Yuan et al., 2024), which can introduce errors due to depth inaccuracies. We therefore adopt the 6-DoF pose trajectory of manipulated objects as our embodiment-agnostic motion representation. We show in Sec. 4 that this results in more accurate imitation than using flow as an intermediary.

**6-DoF Pose Estimation** We consider both model-based and model-free approaches to extract 6-DoF pose object trajectories from human videos. In both approaches, we first obtain an initial mask of the manipulated object using Grounding SAM (Kirillov et al., 2023; Liu et al., 2024). For the model-based setting (meaning that a 3D model of the object is available), we leverage FoundationPose (Wen et al., 2024) to track the object's pose. For the model-free setting – which may be more practical if obtaining a 3D model is not feasible – we adopt a method based on iterative closest point (ICP) (Segal et al., 2009). Specifically, we use

Cutie (Cheng et al., 2024) to propagate the mask across all frames and apply ICP (Segal et al., 2009) to the object point clouds to track relative transformations throughout the video. Finally, we perform pose-graph optimization (Zhou et al., 2018) to ensure consistency across the entire trajectory. In both settings, the pose estimation process produces a sequence of rigid transformations, $\mathcal{T} = (\mathbf{T}_0^c, \mathbf{T}_1^c, \ldots, \mathbf{T}_L^c)$, where $\mathbf{T}_i^c \in SE(3)$ is the 6 DoF pose at frame $i$ with respect to a certain coordinate frame $c$, and $L$ is the length of the video.

### 3.3 Trajectory and Grasp Filtering via Simulation

Now that we have converted the human demonstrations into 6 DoF object pose trajectories, the next step is to execute them on a robot in simulation. This serves two purposes. One is to filter out those that may not be suitable for robot learning. There are two main reasons a trajectory may be unsuitable. First, pose estimation errors can lead to inaccurate trajectories. Second, the extracted trajectory may not be physically achievable by the robot. In either case, it would be harmful to train the robot on such data.

The other purpose of the Simulate step is to generate supervision for *grasping* behavior. Since grasping is highly dependent on end-effector embodiment, it is infeasible for many embodiments to learn grasping by imitating human grasps. However, proper grasping is crucial for robust prehensile manipulation. In order to successfully perform a task, the agent's grasp needs to be both stable and *task-compatible*, meaning that it allows the downstream motion to be performed. Many works overlook the task-compatibility requirement, which is highly important in practice. To illustrate this point, consider the case of turning a door handle clockwise with your right hand (Fig. 2). Using an overhand grip, the motion is easy to execute. In contrast, a firm underhand grip makes the motion difficult and may require additional body movement to compensate. Proper task-compatible grasping is similarly challenging for robots, which are often even less kinematically flexible than humans.

To address this challenge, we propose to filter **grasp-trajectory pairs** and use the data to learn a grasp scoring model as well as a trajectory prediction model. For each video, we first compute the initial object center $\mathbf{u} \in \mathbb{R}^3$ based on the 3D bounding box of the object's point cloud in the first frame. Next, we sample a set of pre-defined anchor grasps, $\hat{\mathcal{G}} = \{\mathbf{G}_k\}_{k=0}^K$, surrounding the object (visualized in Fig. 4). Finally, we execute each $\mathbf{G}_k$, followed by the corresponding trajectory $\mathcal{T}$, and use a waypoint controller in a simulator to verify its feasibility. The robot base in the simulation is placed according to the desired real-world deployment setup. During the simulation, we assume the object becomes rigidly attached to the end-effector when the grasp pose is reached. This is because our goal here is to evaluate task-compatibility without dealing with grasp stability. Accurate simulation of grasp stability would require detailed object information that is generally not feasible to obtain for in-the-wild videos using existing vision methods. At test time, stable grasps will be proposed by an external model and scored based on their nearest anchor grasp (described in Sec. 3.5). This anchor-based design allows for efficient training and inference, and we find that the approximation error it introduces is negligible.

After execution, each grasp-trajectory pair is assigned a binary success label. If all $K$ grasps fail for a given trajectory, we discard the trajectory entirely. In this way, the filtering affects the post-grasp motion learning as well as the grasping. Importantly, the trajectory filtering prevents extreme pose tracking failures from affecting the policy learning. This is a major concern in learning from videos and greatly impacts performance as shown in our experiments.

### 3.4 Policy Learning

We train a simple open-loop policy model using the high-quality grasp-trajectory pairs extracted from human demonstration videos and filtered through simulation. The model takes as input an RGB image (from the start of the video), a binary mask of the target object, and a 2D goal point in pixel coordinates. The 2D goal point serves as a flexible channel for specifying task information (*e.g.*, destination for pick-and-place tasks). We use this input design for simplicity; it is not critical to the overall framework. We use ResNet18 (He et al., 2016) to extract features from the RGB image and the binary mask, and then concatenate them and fuse with the goal point via an MLP. Finally, we adopt separate MLP heads to predict the trajectory and grasp scores ($K$ grasp success probabilities).

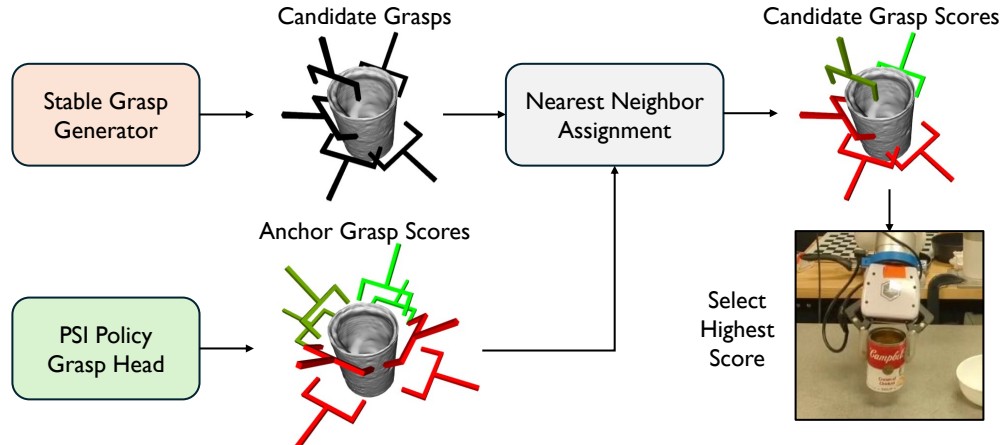

Figure 4: **Modular task-oriented grasping.** PSI exploits existing models for grasp *stability*, while achieving *task-compatibility* via a scoring model trained on simulation data. The scoring model is run first to produce scores for a set of canonical anchor grasps. It can then be combined with any grasp generator in a modular manner by assigning candidate grasps to their nearest anchor grasps.

The model is trained to minimize two losses. The trajectory loss $L_{\mathrm{traj}}$ is an MSE loss on the 6-DoF trajectory. We parameterize the 6-DoF poses as rotation vector and translation vector, and predict poses with respect to an object coordinate frame centered at $\mathbf{u}$. The grasp loss $L_{\mathrm{grasp}}$ is a BCE loss on the success probabilities for the $K$ anchor grasps. We find that a two-stage training scheme works better than full joint training. In the first stage, we only train on $L_{\mathrm{traj}}$. In the second stage, we adopt different training strategies depending on the dataset. For training on HOI4D (Liu et al., 2022b) (1580 episodes), we jointly train on $L_{\mathrm{traj}}$ and $L_{\mathrm{grasp}}$. For training on smaller task-specific datasets (e.g. less than 50 episodes), we freeze all layers except the grasp head and train on $L_{\mathrm{grasp}}$ only. We hypothesize that this performs better because the grasp labels are somewhat noisy and prone to overfitting.

## 3.5 Policy Execution

Although the policy model predicts grasp success probabilities, they only represent the suitability of the grasp *given* that the object is stably picked up. Fortunately, there is a large body of work focusing on the problem of finding task-agnostic stable grasps (Mahler et al., 2017; Sundermeyer et al., 2021; Fang et al., 2023), many of which provide pretrained models that work off-the-shelf. Our model can be combined with any such grasp generator in a simple manner (Fig. 4). Specifically, we first generate a set of candidate 6D grasps, $\mathbf{C}_j$. Each $\mathbf{C}_j$ is then assigned to the closest anchor grasp $\mathbf{G}_k$ based on the magnitude of their rotation difference. We retrieve the success probability predicted by our model for $\mathbf{G}_k$ and assign it to $\mathbf{C}_j$. Finally, the candidate grasp with the highest score is selected for execution. Our method takes advantage of existing models for grasp stability, while achieving task-compatibility through learned selection. Note that it would technically also be possible to evaluate anchor grasp success labels at test time by running simulations with the predicted post-grasp trajectory. However, this would be computationally expensive, so it is desirable to distill the grasp scoring process into a learned model.

# 4 Experiments

## 4.1 Experiment Setup

**Tasks** We evaluate our method with four real-world manipulation tasks: pick-and-place, pouring from a can to a bowl, stirring a pot using a ladle, and drawing on a whiteboard. In each task, there are multiple distractor objects, and the locations of the active object and the distractors are randomized per episode. For pick-and-place, the goal location is randomized as well. Success criteria for each task (used in simulation) are as follows: For pick-and-place, the bottle should end within 15 cm of the table and 8 cm of the final position

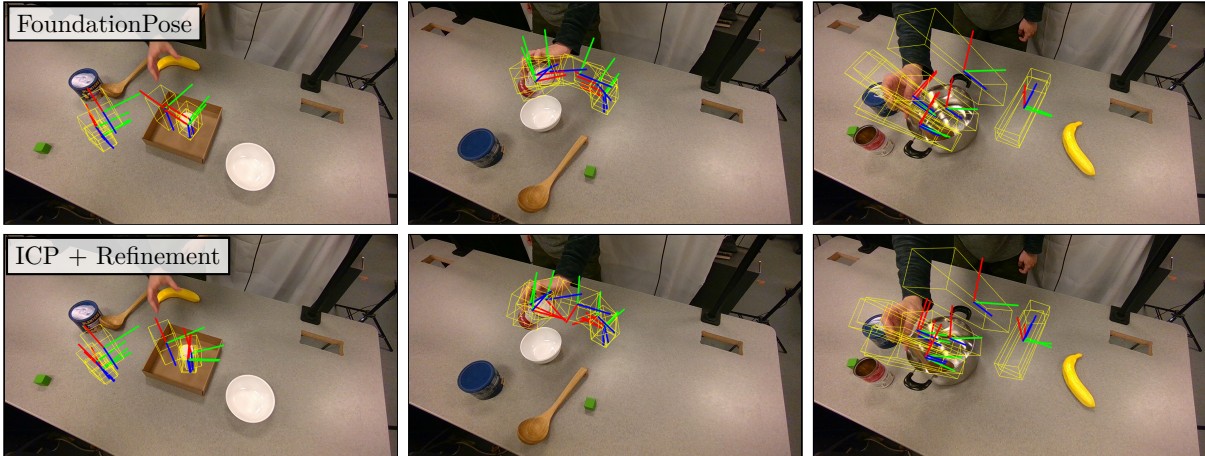

Figure 5: **Qualitative 6D pose tracking results**. We experiment with a model-based method (Foundation-Pose) and a model-free method (ICP + Refinement) for 6D pose tracking. Each RGB image shows the scene at the end of a trajectory. Object bounding boxes transformed using the tracked 6D pose from 8 timesteps are overlaid onto the image. We observe that FoundationPose provides slightly more accurate tracking, but the ICP pipeline still performs satisfactorily for task success.

while remaining within 45 degrees of the upright orientation. For pour, the upwards axis of the can should be more than 60 degrees from the initial orientation, the dot-product with the start-to-goal direction should be positive, and the can should be within 8 cm of the final position. For stir, we define a cylindrical region of height 8 cm and radius 15 cm, and require that the object move along a path of more than 10 cm within the region. For draw, it is the same but with height 5 cm, radius 12 cm, and path length 20 cm.

**Training data**   We collect 50 human video demonstrations for each task. The videos are captured using a fixed Intel Realsense D455 RGB-D camera. We use 35 demonstrations for training, and 15 for validation. For pick-and-place, the 2D goals are calculated by projecting the final object center. For pretraining with PSI on HOI4D, we use 1580 pick-and-place clips, containing all object categories except "Storage Furniture", "Safe", "Trash Can", and "Chair". The clips are cut using the "reachout" and "putdown" event labels. All of the policy models are fine-tuned from the same HOI4D-pretrained checkpoint unless otherwise specified.

**Robot evaluation**   For real-world robot evaluation, we use a UFACTORY xArm7 with a UFACTORY xArm gripper, mounted next to the table viewed by the D455 camera. The camera is fixed in the same place as during training, and the camera-to-base transform is calibrated via hand-eye calibration. For the pick-and-place, the goal is set interactively by clicking a point on the observed RGB image. To generate task-agnostic candidate grasps, we use a heuristic for each object. For the pick-and-place bottle and the can, the candidate grasps follow the same grid as the anchor grasps. For the ladle, we set the grasp point by finding the 3D point on the ladle closest to the camera, and then grasp either towards or away from the robot along the $y$ direction. Empirically, these methods reliably produce stable grasps on each object.

**Implementation details**   For pose graph optimization, we form edges between pairs 32 and 64 frames apart, and use default Open3D settings otherwise. For simulation, we leverage the operational space controller in robosuite (Zhu et al., 2020). We use $K = 8$ pre-defined anchor grasps, covering the four cardinal directions with respect to the robot base and elevation angles of 10 degrees and 50 degrees. We find that this provides sufficient coverage for accurate nearest-neighbor grasp scoring – further densifying the grid does not significantly improve grasp selection. For pose tracking method details, as well as policy model architecture and optimizer details, please see Appendix C.

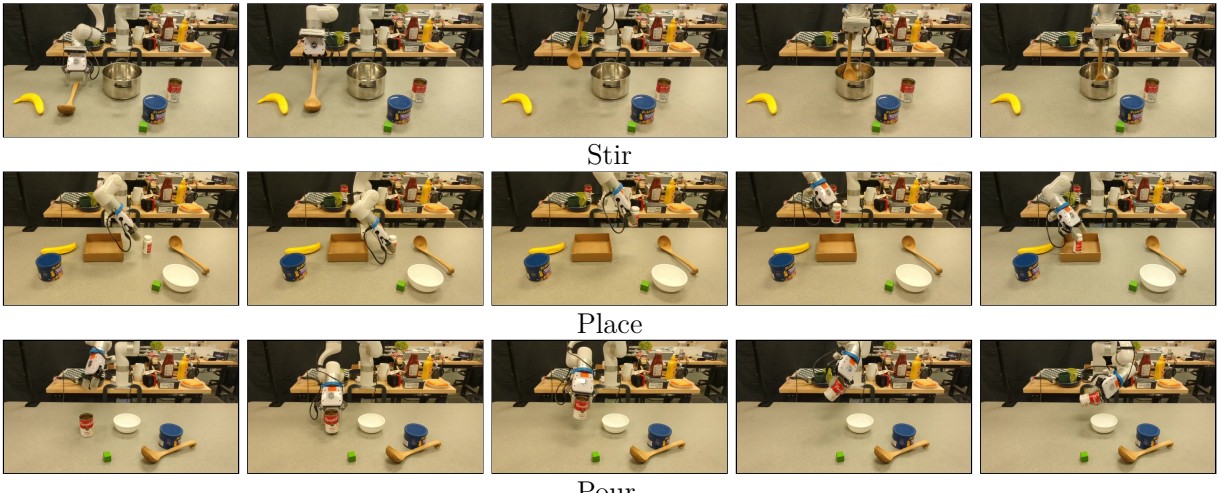

Stir

Place

Pour

Figure 6: **Real-world execution**. We show sequences of real-world robot execution of our learned policies. Each policy is trained using only 35 human demonstrations, which takes less than one hour to collect. We observe that our method is able to learn relatively complex motions such as stirring the pot with the ladle.

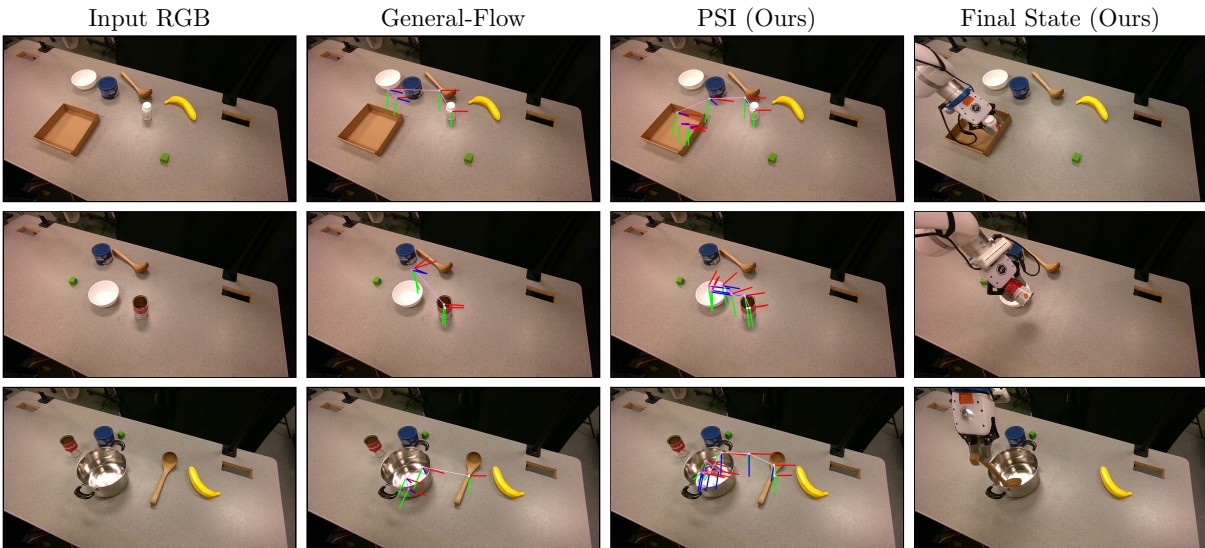

Figure 7: **Qualitative policy prediction results**. We compare the $SE(3)$ transformations predicted by General-Flow (Yuan et al., 2024) and our method (on unseen images). Both methods are trained on the same underlying motions extracted by FoundationPose. The trajectories produced by our model are significantly more accurate to the desired tasks. The right column shows the end result of executing the trajectory after grasping using our method.

## 4.2 Pose Tracking Quality

We first validate the accuracy of our 6D pose tracking pipelines qualitatively in Fig. 5. We find that both the FoundationPose (model-based) and the ICP-based (model-free) pipeline perform reasonably for most of the videos, providing pose trajectories that correspond to success with respect to the manipulation task. FoundationPose is slightly more accurate overall. ICP struggles more when there is significant occlusion of the object, such as when the hand occludes the handle of the ladle. Both methods encounter difficulties with approximately symmetric objects such as the bottle and the can, and they may produce rotations that drift around the axis of symmetry. These shortcomings motivate the need for trajectory filtering.

Table 1: Manipulation success rate with and without trajectory filtering and task-compatible grasp selection. FP (FoundationPose) and ICP indicate which 6D pose tracking method was used. **Bold**: best method in each setting.

| Method | P&P | Pour | Stir | Draw |
|---|---|---|---|---|
| No trajectory filtering (FP) | 6/20 | 12/20 | 16/20 | **12/20** |
| Naive grasp (FP) | 5/20 | 8/20 | 10/20 | 1/20 |
| Ours (FP) | **16/20** | **13/20** | **20/20** | **12/20** |
| No trajectory filtering (ICP) | 10/20 | 8/20 | 8/20 | 0/20 |
| Naive grasp (ICP) | 4/20 | 7/20 | 11/20 | 0/20 |
| Ours (ICP) | **15/20** | **13/20** | **18/20** | 0/20 |

Table 2: Comparison between flow prediction and direct 6D pose prediction. The flow predictions are converted to end-effector actions by solving a least-squares problem. **Bold**: best method in each setting.

| Method | P&P | Pour | Stir | Draw |
|---|---|---|---|---|
| General-Flow (Yuan et al., 2024) (FP) | 7/20 | 4/20 | 1/20 | 0/20 |
| Ours (FP) | **16/20** | **13/20** | **20/20** | **12/20** |
| General-Flow (Yuan et al., 2024) (ICP) | 5/20 | 0/20 | 0/20 | 0/20 |
| Ours (ICP) | **15/20** | **13/20** | **18/20** | 0/20 |

Table 3: Comparison of pretraining methods. Our method entails running the PSI pipeline on pick-and-place clips from HOI4D (Liu et al., 2022b). **Bold**: best method.

| Method | P&P | Pour | Stir | Draw |
|---|---|---|---|---|
| ImageNet (Deng et al., 2009) | 10/20 | 11/20 | 6/20 | 0/20 |
| R3M (Nair et al., 2023) | 5/20 | **13/20** | 10/20 | 0/20 |
| Ours | **16/20** | **13/20** | **20/20** | **12/20** |

## 4.3 Effect of Simulation-Based Filtering

We conduct ablative experiments to study the effect of trajectory filtering (discarding pose trajectories that fail with any grasp) and task-oriented grasping. The results are shown in Tab. 1. For each task, success rate is evaluated over 20 episodes with novel starting states. Without trajectory filtering, the training data from FoundationPose and ICP both contain some highly erroneous pose tracking results, degrading the performance of the policy. Note that, without filtering, ICP generally results in a significantly worse policy than FP. With filtering, the gap is largely closed, except on the Draw task in which the small size of the marker gives too few 3D points for ICP to track well.

To measure the advantage of task-oriented grasping over task-agnostic grasping, we replace our policy-selected grasps with random candidate grasps. This causes many failures due to the desired trajectory being impossible to achieve when starting from a sub-optimal grasp. The large performance drop highlights the need for task-oriented grasping in prehensile manipulation.

We also record the number of discarded pose trajectories for each task. For FoundationPose, the numbers are (out of 50 trajectories): 6 for P&P, 5 for Pour, 3 for Stir, 3 for Draw. For ICP, they are: 4 for P&P, 2 for Pour, 3 for Stir, 3 for Draw. The primary cause of these is poor pose tracking due to either motion blur or occlusion of the object. We recommend the robot learning community to pay extra attention to these factors during data collection.

Table 4: Simulation-based evaluation of PSI policies (with real inputs) across different robot embodiments.

| Robot | P&P | Pour | Stir | Draw |
|---|---|---|---|---|
| xArm7 | 11/13 | 10/14 | 13/13 | 13/14 |
| Franka Panda | 11/13 | 6/14 | 13/13 | 12/14 |
| Kinova Gen3 | 10/13 | 7/14 | 13/13 | 13/14 |
| UR5e | 11/13 | 11/14 | 13/13 | 10/14 |

### 4.4 Flow vs. 6D Pose Prediction

To study the effectiveness of using object flow versus 6D pose as a learning target, we compare our framework with General-Flow (Yuan et al., 2024), which predicts 3D flow from point cloud input and then computes $SE(3)$ end-effector transformations using singular-value decomposition (Besl & McKay, 1992). Since General-Flow provides only the post-grasp actions and thus relies on external grasp generation, we combine it with our method's grasping so that the evaluation focuses on post-grasp trajectory quality. We use the official HOI4D-pretrained ScaleFlow-B model and fine-tune it on the same training demonstrations as our method. To obtain flow labels, we take 3D points on each active object and transform them based on our extracted 6D pose for consistency. We compare the robot execution results in Tab. 2. We find that our method performs better across all the tasks, both in the model-based and the model-free setting. We compare the policy predictions qualitatively in Fig. 7. Overall, the $SE(3)$ transformations obtained by predicting flow and solving for alignment are less accurate than those obtained by direct prediction. Full real-world rollouts on the xArm7 are shown in Fig. 6.

### 4.5 Pretraining Comparison

We compare the effectiveness of pretraining with PSI on HOI4D versus other pretraining methods in Tab. 3. ImageNet refers to using the default ImageNet-1K (Deng et al., 2009) weights for the ResNet18 RGB encoder, and R3M (Nair et al., 2023) trains manipulation-oriented features through contrastive learning on Ego4D (Grauman et al., 2022). All of the methods are trained with FoundationPose-based data. We find that pretraining with PSI on HOI4D provides large benefits on most of the tasks, but less so on the pour task. This makes sense because the pour task is heavily rotation-focused, and the model is not likely to learn much about rotational movements from P&P data in HOI4D.

### 4.6 Robot Embodiment Comparison

PSI can be applied to learn manipulation policies for various robot embodiments. We validate the generality of the framework by evaluating policies for different robot arms in simulation. Specifically, we run the Simulate and Imitate steps for xArm7, Franka Panda, Kinova Gen3, and UR5e robots, and evaluate by running model inference on real-world validation observations and then executing the planned actions in simulation. The validation trajectories were filtered (based on xArm7 feasibility) in order to ensure accurate success evaluation. During the simulated execution, we assume that the gripper rigidly attaches to the object upon reaching the grasp pose (as in our Simulate step), abstracting away the simulation of grasp stability. The results are shown in Tab. 4. PSI successfully trains policies for each of the different robot embodiments, demonstrating its versatility. We note that the pour task leads to the greatest variance across embodiments. This makes sense, as the large rotations required by the pouring motion frequently challenge kinematic limits, making them feasible for certain robots but not others.

## 5 Limitations

One limitation of our framework is that the use of 6-DoF pose restricts us to videos in which the grasped object is rigid or approximately rigid. The motion of articulated and deformable objects may not be adequately represented by the 6DoF representation. Another limitation is that the current framework would encounter a visual domain gap if applied to learn closed-loop policies. Currently, the model only observes the initial frame of each video, in which there is an unobstructed view of the scene. However, intermediate frames of human

hand-object interaction videos often involve a significant degree of occlusion by the human body, and training on such visual inputs will result in a domain gap during robot deployment. Fortunately, inpainting and insertion rendering methods (Bahl et al., 2022; Liu et al., 2022a; Lepert et al., 2025a) have shown potential to remedy this issue, and they can be combined with our framework.

## 6 Conclusion

We presented Perceive-Simulate-Imitate (PSI), a framework for cross-embodiment visual imitation. PSI extracts embodiment-agnostic 6DoF pose trajectories and leverages simulation for filtering grasp-trajectory pairs. This filtering process not only removes infeasible trajectories, but also enables learning of task-oriented grasping behavior. In our experiments, we show that PSI allows for sample-efficient learning of real-world manipulation tasks. So far, we have mainly focused on task-specific training, aiming to demonstrate that precise manipulation policies can be trained quickly and cheaply with human videos only. An important direction for future work is to explore how the trajectory and grasp labels produced by PSI can be used for larger-scale training of generalist foundation models. We also anticipate that PSI can be extended with more advanced Real2Sim techniques in the future to further improve the quality of the extracted training signals.

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

## A    Simulation Filtering Visualization

We visualize example failure rollouts in Fig. 8 to illustrate what grasp-trajectory pairs are filtered in the Simulate step. A common occurrence is the robot grasping the object from a direction in which it needs to later rotate the object towards (Fig. 8, left), causing its wrist link to collide with the table. Note that this is not a failure of the waypoint controller – there is indeed no way to follow the desired trajectory without colliding, due to the morphology of the robot. Executing task-compatible grasps is necessary to avoid such issues. The other panel shows a trajectory that resulted from a large pose tracking failure. When such erroneous trajectories are mixed into the training data, the policy's predictions strongly degrade.

## B    Data Scaling Analysis

We examine how the performance of PSI policies varies with the number of training demonstrations in Tab. 5, again using simulation-based proxy evaluation. Generally, the manipulation performance steadily increases in this data regime. For the Stir task, the model is able to achieve full success rate with only 15 demonstrations. This may be due to the task's initial state distribution being narrower, though we also note that the success criteria for the Stir task in simulation may be much more lenient than in the real-world, because the simulation does not factor in collisions with the pot (which are the cause of a large portion of real-world failures).

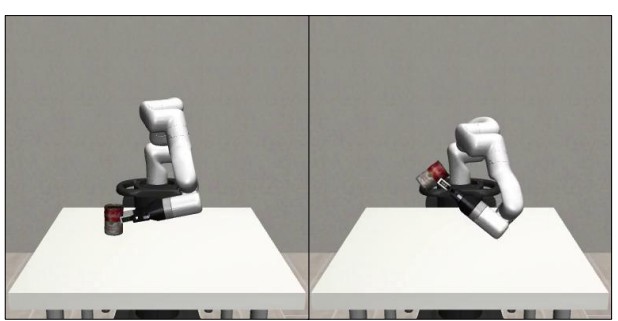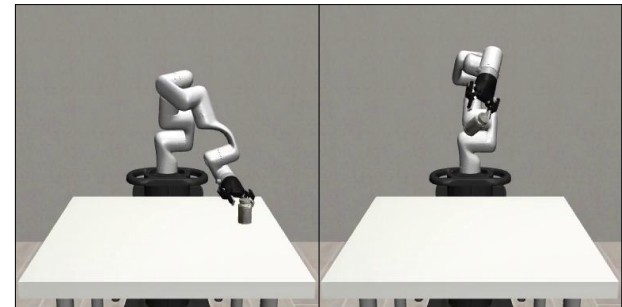

Figure 8: **Different filtered grasp-trajectory pairs.** We visualize two cases of a grasp-trajectory pair being filtered in simulation. In the first case, the robot cannot complete the trajectory because the arm collides with the table (due to a poor grasp). In the second case, the trajectory is erroneous due to a pose tracking failure.

Table 5: Simulation-based evaluation of PSI policies trained with different amounts of demonstrations. The inputs to the policies are held-out real-world observations.

| # of training demos | P&P | Pour | Stir | Draw |
|---|---|---|---|---|
| 15 | 7/13 | 5/14 | 13/13 | 10/14 |
| 25 | 9/13 | 9/14 | 13/13 | 12/14 |
| 35 | 11/13 | 10/14 | 13/13 | 12/14 |

## C  Additional Experiment Details

**3D object center calculation**   A number of processing steps are used to compute the 3D object centers from the RGB-D data robustly. Firstly, we erode the segmentation mask using a $3 \times 3$ kernel to remove border pixels from the object. The object points are unprojected to 3D using the camera intrinsics. Then, the bounding box is determined by taking the 5th and 95th percentile points in each dimension, and its center is taken as the final center.

**ICP implementation**   Similarly, for ICP pose tracking, a few preprocessing steps are used to clean up the point clouds. We erode the segmentation mask using a $5 \times 5$ kernel and perform statistical outlier removal using Open3D Zhou et al. (2018) with 30 neighbors and a standard deviation ratio of 2.0. The video frames are processed in order, and frames in which there are less than 500 object points are skipped, since heavily occluded point clouds generally result in poor pairwise registration. The ICP registration (Open3D Generalized ICP) is run first with a coarse correspondence distance of 8 cm and then a fine correspondence distance of 2 cm. When the computed transformation between one frame and the next has an object-center translation greater than 2 cm or a rotation magnitude greater than 0.2 radians, it is assumed to be erroneous and set to the previous frame's transformation.

**3D object model acquisition**   For FoundationPose model-based pose tracking, 3D models of the target object are required. We obtained these by scanning each object using the Polycam app with a Google Pixel 6. We run FoundationPose with 8 refinement iterations for pose estimation and 2 refinement iterations for tracking.

**Network architecture and optimizer**   For our policy model, the goal point embedding has dimension 32, and the MLP fuser has two layers of size 512. The trajectory decoder has two layers of size 128, and predicts 16 waypoints. The grasp decoder has two layers of size 64. For training, we use an AdamW optimizer with a learning rate of 0.0001 and batch size of 64 (full dataset for our task-specific data). For pretraining on HOI4D, we train on $L_{\text{traj}}$ for 800 epochs and then on $L_{\text{traj}} + 0.01 L_{\text{grasp}}$ for 200 epochs. For our task-specific data, we train on $L_{\text{traj}}$ for 1000 epochs and then on $L_{\text{grasp}}$ with $0.1\times$ learning rate for 500 epochs.

## D   Simulation Runtime Analysis

We measured the average time needed to simulate the execution of one grasp-trajectory pair (in robosuite (Zhu et al., 2020)) and found it to be 9.68 seconds. The simulations of different grasp-trajectory pairs can be cleanly parallelized and thus the total time directly depends on the number of threads available for parallelization. For example, if 32 threads are available, processing a 50-demonstration dataset for $K = 8$ anchor grasps would take $\frac{50 \times 8 \times 9.68}{32} = 121$ seconds. This runtime could potentially be further reduced by leveraging GPU-accelerated simulators such as IsaacSim (NVIDIA). Methods such as Track2Act (Bharadhwaj et al., 2024b) and EgoMimic (Kareer et al., 2025) rely on an entire additional cycle of demonstration data collection and model training in order to learn task-oriented grasping capabilities. Note also that the data must be robot action demonstrations, requiring a full teleoperation setup. Our method replaces this entire process with running a few minutes of simulation, which is far less costly and does not require any robot hardware.

