# OpenReview forum: "Imitating What Works: Simulation-Filtered Modular Policy Learning from Human Videos"
_TMLR — Accepted by TMLR_

### Review · Reviewer_xEG2 · 2026-03-13

**Summary Of Contributions:**

This paper studies modular methods for prehensile manipulation, where tasks are split into grasping and post-grasp motions. The authors primarily focus on the grasping phase, arguing that grasping must be task-aware, as not all grasp poses lead to successful task execution. To address this, they propose a simulation-based approach to filter out low-quality demonstrations and grasp poses that result in failure. The proposed framework, PSI, utilizes a computer vision pipeline to extract object positions and trajectories from human videos. This data is then passed through the simulation-based filter to curate a training dataset, which is used to train a policy network via behavior cloning (supervised learning). Experiments across four manipulation tasks showcase the effectiveness of the proposed method.

### Strength
- The paper is generally well-written.
- The method is clearly motivated and mostly well-explained.
- The experimental results support the authors' claims regarding the importance of task-oriented grasping and demonstrate the utility of the proposed simulation-based filter.

### Weakness
I am unconvinced that this paper is the right fit for TMLR, as it is heavily robotics-oriented with limited focus on machine learning methodologies. The core problem addressed is how to generate task-dependent grasp poses, and the authors' primary contribution to solving this is the use of simulation. The only machine learning component in the pipeline is the policy model training, which relies on standard behavior cloning. Therefore, I believe this work is better suited for a traditional robotics venue like ICRA or IROS, or a hybrid venue like CoRL, rather than a machine learning venue like TMLR.

**Audience:**

No

**Audience Explanation:**

As noted in the summary, I believe the paper is a better fit for a robotics venue than a machine learning one, as its primary contributions lie outside the scope of machine learning.

**Claims And Evidence:**

Yes

**Claims Explanation:**

The authors' main claim is that task-dependent grasping is important and that the proposed simulation-based filter is effective. This claim is well-supported by the experimental results, particularly Table 1.

**Requested Changes:**

- The authors should better articulate how their methods and findings contribute to the broader machine learning community, rather than solely the robotics community.
- Additionally, several technical details remain unclear, and I would appreciate it if the authors could elaborate on the following points:
	- The authors mention training the policy to output scores for a set of pre-defined anchor grasping poses. Consequently, during evaluation, the generated grasp pose must be mapped to the nearest anchor. Why not train the policy network to directly accept the grasp pose from the generator and output a score for any arbitrary pose?
	- Because the proposed method relies on simulation to label demonstrations, it is worth detailing how the simulation environment is initialized based on the demonstration data. I think the success of a grasp pose can be heavily dependent on the position of the robot base. For instance, if an object and its post-grasp trajectory lie on the boundary of the robot's workspace, a random grasp is less likely to succeed, as the robot can easily reach a kinematic singularity. Conversely, if the object and trajectories are near the center of the workspace, almost any random grasp might result in a successful task execution.
	- The authors state that 50 trajectories were collected for each task, but there is little detail regarding how they were gathered. It would be highly beneficial to document the variety within these demonstrations (e.g., are they from the same scene or different scenes? What variables change from one demonstration to the next?). Most importantly, what is the extent of the difference between the demonstration settings and the evaluation settings?
- The authors should fix the citation format, use \citep for the parenthetical citations, while \citet for inline citations.

---

> ### Author Response · Authors · 2026-04-14
>
> Thank you for your feedback on our paper; we really appreciate it. We provide itemized responses to your concerns and questions below:
>
> - **Relevance to TMLR**: We respectfully disagree that our paper is out-of-scope for TMLR. A significant portion of TMLR’s audience are researchers in robotics or adjacent areas, and numerous robot learning papers have previously been published in TMLR. The acceptance criterion specifically asks “would *at least some* individuals in TMLR’s audience be interested in the paper”. Considering that the topic of learning from human videos is of high interest to many robot learning researchers, we believe that our paper clearly satisfies this criterion. Additionally, note that our contribution is not simply the idea of using simulation; the policy model must be newly designed to learn grasp-trajectory compatibility from the simulation results. Specifically, our model is modified to predict anchor grasp scores jointly with end-effector trajectories in order to enable task-compatible integration with an external grasp generator. Thus, the machine learning methodology here is different from standard behavior cloning.
> - **Why not train the policy network to output a score for any arbitrary pose**: This design is also possible, but it would require running the model again for every candidate grasp, which is not efficient if we want to evaluate a large number of candidate grasps. Our anchor-based design only requires running the scoring model once, which is much more efficient while still providing adequate accuracy as shown in our experiments. The subsequent nearest-neighbor mapping process has negligible computational cost. We will add more explanation in Sec. 3.3 about this design choice.
> - **Simulation environment initialization**: You are right that the grasp success depends on the robot base position. However, this is not a problem in our case, because the simulation can be set to match the actual robot deployment setup. Pose tracking methods give trajectory data in the camera coordinate frame. Using the desired camera-to-robot transform (which can be obtained for the real setup using standard camera calibration), we can place the simulated robot so that it is consistent with the deployment setup. We will update Sec. 3.3 to explain this more clearly.
> - **Task details**: We provided a brief explanation of the variability within each task in Sec. 4.1. (In each task, there are multiple distractor objects, and the locations of the active object and the distractors are randomized. For pick-and-place, the goal location is randomized as well.) The actual objects and setup can be seen in Figs. 5, 6, and 7. The demonstration and evaluation settings both follow the aforementioned randomizations. Currently, the background scene is the same throughout the demonstrations and evaluation; we did not investigate generalization to new scenes. However, note that HOI4D pretraining does involve a large variety of scenes.
> - **Citation format**: Thanks for pointing this out; we will be sure to fix this.
>
> We will upload a revised PDF with the described changes shortly. Please feel free to follow up on this thread in the meantime!

---

### Review · Reviewer_VPAz · 2026-03-15

**Summary Of Contributions:**

This work proposes PSI, a framework for prehensile manipulation from human videos. The work argues that whilst using stable grasp generators is a good way forward, adopting grasp generators naively will likely generate task-agnostic grasping suggestions, which may be harmful for robot learning. The main key of the proposed method to address this challenge is to generate task-oriented grasping by utilising simulations to filter out unusable grasp suggestions.

---
### **Strengths**
**(S1. readability)** This work is well written and the flow is easily readable.
**(S2. importance)** The motivation of the work and the challenge this work tries to address is clear and intuitive. Bridging the gap between grasping suggestions from generators and the actual feasible grasp is an important contribution.
**(S3. effectiveness)** The results are impressive, especially for the cross-embodiment, showing embodiment-agnostic results.

---
### **Weaknesses**
**(W1. heuristics)** There are several heuristic statements, for example, at the bottom of page 5, "if a 3D model is available, … is unavailable"; or at the end of page 6, "In the second stage, if the amount of data is large, …, If it is small, …". Whilst I think this is fine, these types of statements make it hard to adapt to future tasks not experimented in this work. Furthermore, part of the experiments are to justify these choices, such as using FP instead of ICP, or using HOI4D instead of R3M/ImageNet. It is a bit confusing at first which "default" method is included in the proposed PSI framework.
**(W2. presentation)** Results can be presented with more care and clarity. For example, in the Tables 1, 2 and 3 it’s written $x/20$. One would assume it means 20 tries, but 20 tries with which objects? In the works it was written that symmetric objects can affect pose estimation and therefore the final performance, the objects used in these experiment tasks should be noted. For Table 4 too, where $x/14$ is written.
**(W3. computation costs)** The work’s main contribution is filtered grasp suggestions using simulation. Since this introduces calculation overhead due to simulations, it would be welcomed if there were some comparisons on the computation costs of different types of approaches, especially for the additional simulation step required for filtering versus other approaches that extracts task-specific information.  I think this is probably the most important weakness of this work for me and should be discussed more.
**(W4. task-compatible grasps)** To my understanding, the authors discard a trajectory only if K grasps all fail in the simulation. This shows the (additional) importance of a “good” predefined K anchor grasps. Currently there is no discussion on how K is chosen, nor its effect. Some discussion on predefined K would be appreciated.

---
### **Questions**
**(Q1. comparison)** From the experiments it reads that FP is the superior approach to ICP+refinement after combining with the proposed PSI. Given we can somewhat obtain a 3D object model quite easily (as noted by the authors by using a google pixel), what are the disadvantages of FP compared to ICP in the authors opinion regarding prehensile manipulation (i.e. why not just always use FP for the proposed method?)?
**(Q2. data usage)** I’m a bit confused on how HOI4D data and the robot demonstrations that the authors collected are used. To my understanding, all tasks pre-trains using HOI4D data and then use the collected demonstrations for fine-tuning?
**(Q3. training flow)** I might have misunderstood somewhere but, do the validation trajectories also get discarded trajectories? Furthermore, during pre-training with HOI4D data, I would assume that no trajectory filtering is done?

---
### **Minor Comments**
**(M1. references)** Figure 1 and Figure 3 are not described nor referred anywhere in the texts.
**(M2. missing hyperparameters)** For the number of the predefined K anchor grasps, I cannot seem to find the actual K used written clearly. Is it 4 directions * 2 degrees = 8 predefined grasps?
**(M3. appendix)** Maybe more of a personal choice, but whilst reading this work sometimes I feel that several details, such as networks architecture, learning rate and optimiser, can be moved to the appendix to improve reading flow.

**Audience:**

Yes

**Audience Explanation:**

The proposed filtering by using simulation scheme fill a gap in the literature regarding learning only from human videos. The proposed filtering scheme neat way to get task-relevant grasp suggestions, is effective and generalised well to different embodiments.

**Broader Impact Concerns:**

There are no specific concerns as far as I am aware.

**Claims And Evidence:**

Yes

**Claims Explanation:**

There are clear evidence that the proposed method is effective in the current form. Grasp suggestions from external grasp generators can  be noisy and unsuitable for the specific tasks to solve, and the proposed method improves on this by adding a simulation filter layer that only task-compatible grasps are kept. This greatly improves the robustness of grasps in the experiments. There are some unclear part that can further improve the work but if addressed properly, the proposed work would be a good addition to the field.

**Requested Changes:**

**(R1. ablation on computation costs)** Address W3 by having a brief comparison between computation cost of different types of approaches regarding obtaining task-compatible trajectories.
**(R2. clarification)** Address W2 by adding more details about the experiments presentation such as .
**(R3. clarification)** Address M2 by adding the hyperparameters (could also address M3 as well, authors’ choice).
**(R4. clarification)** Address Q1 by having a brief description of pros/cons of FP v.s. ICP, given that FP seems to always perform better for PSI.
**(R5. clarification)** Address Q2 by adding descriptions regarding data usage.
**(R6. clarification)** Address Q3 in the appropriate part of the texts regarding training flow.
**(R7. missing descriptions and references)** Address M1 by adding descriptions regarding Figure 1 and Figure 3 in the texts.
**(R8. ablation, optional)** If possible, address W4 by adding discussion on the predefined anchor grasps affecting final performance. Also related to M2.

---

> ### Author Response · Authors · 2026-04-14
>
> Thank you for your feedback on our paper; we really appreciate it. We provide itemized responses to your concerns and questions below:
>
> - **(R1) Computational cost of simulation**: We measured the average time needed to simulate one grasp-trajectory pair and found it to be $9.68$ seconds. The simulations of different grasp-trajectory pairs can be cleanly parallelized and thus the total time directly depends on the number of threads available for parallelization. For example, if 32 threads are available, processing a 50-demonstration dataset would take $\frac{50 \times 8 \times 9.68} {32} = 121$ seconds. Methods such as Track2Act rely on an entire additional cycle of demonstration data collection and model training in order to learn grasping capabilities. Note also that the data must be robot action demonstrations, requiring a full teleoperation setup. Our method replaces this entire process with running a few minutes of simulation. The cost of the demonstration data collection could depend on many factors, so it is not directly comparable, but it is definitely more costly than running the simulation. We will add a section to the appendix discussing these points.
> - **(R2) Experiments clarification**: The x/20 refers to the success rate over 20 episodes with different starting states. The format follows standard practice in robot learning evaluations. The object-to-be-manipulated is fixed per task, but its location varies, and there are randomized distractor objects as well (described in Sec. 4.1). The policy needs to generalize over these factors to achieve a high success rate. We will make the format of the results more clear by adding explanations in Sec. 4.3. The particular objects used can be seen in Figs. 5, 6, and 7. We note that most of the objects may exhibit symmetry-induced pose ambiguities from some viewpoints but not others. For example, the Campbell’s can has rotational ambiguity when viewed from the top, but not from the side where the printed wrapper can be seen.
> - **(R3) Value of K**: Yes, K = 8 in the experiments. We will make this explicit in the implementation details.
> - **(R3) Moving details to appendix**: We are on board with your suggestion and will move the network architecture and optimizer details to the appendix.
> - **(R4) Pros/cons of FP vs. ICP**: FP gives more accurate pose tracking, but it requires having access to a complete 3D model of the object a priori. Although scanning objects is possible using software such as Polycam, it is still a somewhat laborious process taking 10-15 minutes per object. ICP and other model-free methods do not require any 3D model, making them more scalable in practice. We will add an explanation about this in Sec. 4.2.
> - **(R5) HOI4D usage**: Your understanding is correct: HOI4D is used for pre-training and then the models for each task are fine-tuned from that checkpoint. The exception is Table 3, in which we ablate the HOI4D pretraining to compare with other pretrained methods. We will update Sec. 4.1 to explain this more clearly.
> - **(R6) Validation filtering**: Discarded trajectories are excluded from the validation set, as they will almost certainly result in failure and thus do not give useful evaluation information. This is the reason that the results in Table 4 involve differing numbers of episodes. We will update Sec. 4.6 to clarify this.
> - **(R6) HOI4D filtering**: The HOI4D pretraining data is also filtered using our simulation methodology. Overall, the pretraining is done using the same Perceive-Simulate-Imitate framework as the task-specific training, showing the framework’s versatility.  We will update Sec. 4.1 to clarify this.
> - **(R7) Figure references**: Thanks for pointing this out. We will add references to Fig. 1 in Sec. 1 and to Fig. 3 in Sec. 3.1.
>
> (continued below)

---

> > ### Author Response · Authors · 2026-04-14
> >
> > - **(R8) Effect of anchor grasp choice**: Intuitively, what we want is for the anchor grasps to cover the space of possible grasps so that nearest-neighbor approximation is accurate. Increasing K to densify the anchor set should make the grasp scoring more accurate, but it comes at the cost of requiring more simulations. We find that our grid design with K=8 provides sufficient coverage and that further densifying the anchor set does not significantly improve performance. We will add discussion about this point in Sec. 4.1.
> > - **(W1) Heuristic statements**: Regarding model-based vs. model-free tracking, these should not be considered as design choices in our method, but rather two different problem settings that may be encountered in practice. We envision that practitioners may choose to pursue either a model-based or model-free pipeline based on how convenient it is to obtain 3D object models for their use case. We will change the wording in Sec. 3.2 to clarify this. Regarding the training scheme for large vs. small datasets, it is indeed true that the combination of losses may need to be tuned per dataset for optimal results. We will update Sec. 3.4 to clarify this.
> >
> > We will upload a revised PDF with the described changes shortly. Please feel free to follow up on this thread in the meantime!

---

### Review · Reviewer_RjaJ · 2026-04-10

**Summary Of Contributions:**

This paper tries to tackle such a problem: learning robot prehensile manipulation directly from human videos is difficult due to the embodiment gap. While existing modular approaches attempt to solve this by using external grasp generators, they often fail because they select "task-agnostic" grasps. These grasps might be physically stable, but they prevent the robot from completing the necessary downstream motion to finish the task.

The solution proposed by this paper is a three-step framework:
- Perceive: The system uses 3D vision techniques to extract 6-DoF object pose trajectories from human demonstration videos.
- Simulate: It tests these trajectories alongside various candidate grasps in a physics simulator to filter out impossible or erroneous motions and generate success labels for task-compatible grasps.
- Imitate: It uses behavioral cloning to train a policy on this filtered data, teaching the model to predict both post-grasp motions and grasp suitability scores.

This PSI framework successfully trains robots to perform complex, real-world tasks using zero actual robot demonstration data. The authors note that the simulation-based filtering step is critical to this success; without it, unachievable trajectories and task-incompatible grasps result in high failure rates.


## Key Strengths
- The simulation filtering step effectively identifies and discards pose trajectories that are either inaccurate due to tracking errors or physically unachievable for the robot.
- The approach is pretty sample-efficient, demonstrating the ability to learn relatively complex real-world motions like stirring a pot using a policy trained on just 35 human demonstrations.

## Key Weaknesses
- Training a purely open-loop policy that predicts an entire trajectory from a single initial RGB image makes the real-world execution brittle and cannot recover from minor errors.
- Using a 2D pixel coordinate goal point to specify task intent is a highly limiting interface.
- While the paper heavily motivates the method as a way to unlock "highly scalable data", the empirical validation relies on training narrow, task-specific policies using only 35 demonstrations each.
- The reliance on 6-DoF object poses limits the framework to handle articulated and deformable objects.
- The model-free pose tracking pipeline (ICP) has difficulties tracking objects when there is significant occlusion, such as a hand covering a ladle's handle.

**Audience:**

Yes

**Audience Explanation:**

Yes, I think AI and robotics researchers are always motivated to find scalable ways to teach robots directly from human videos, as collecting real-world robot data is expensive and difficult. This paper would be interesting to them because it offers a practical solution to the embodiment gap.

**Claims And Evidence:**

Yes

**Claims Explanation:**

I think the claims in the submission are largely supported by clear empirical evidence, though the scope of that evidence is somewhat limited by small sample sizes and simulation simplifications.

The authors effectively isolate the contributions of their core components. And this paper also provides quantitative, real-world comparisons against existing methods. For example, they demonstrate that their direct 6-DoF pose prediction yields higher success rates than predicting flow and converting it to actions (General-Flow). The central claim that a policy can be learned purely from human videos without robot data, is validated through actual physical execution on an xArm7 robot.

However, the claims of robust manipulation are based on a very constrained dataset, utilizing only 35 human demonstrations for training and evaluating on just 20 trials per task in the real world. In addition, while the authors claim the framework can generalize to different robot embodiments, the evidence for this specific claim relies entirely on executing planned actions in simulation rather than physical hardware testing.

**Requested Changes:**

N/A

---

> ### Author Response · Authors · 2026-04-14
>
> Thank you for your feedback on our paper; we really appreciate it. We provide itemized responses to your concerns and questions below:
>
> - **Open-loop policy**: We acknowledge that the current framework does not handle closed-loop policy training, and we discuss this in Sec. 5. The primary reason for the limitation is the visual domain gap induced by body occlusion. Inpainting and insertion rendering methods provide a potential solution by replacing the human bodies with robot bodies. Such methods have made rapid progress in recent years, and they can be combined with our framework to close the gap. That being said, it is possible to perform well on many real tasks using open-loop policies. Our experiments are evidence of this.
> - **2D goal point interface**: We do not intend the 2D goal point design to be part of our contribution; it is simply a mechanism to accommodate evaluating our framework on a pick-and-place task with variable goals. We will update Sec. 3.4 to clarify this.
> - **Task-specific setting**: The approach of learning from human videos is inherently less expensive and more scalable than the alternative of learning from teleoperated robot demonstrations, regardless of whether the goal is task-specific learning or multi-task learning. The difficulty lies in actually producing high-quality policies despite the embodiment gap. We show that our framework can achieve this. Our experiments are primarily in the task-specific learning setting, but we also demonstrate the potential for pretraining using data from HOI4D in Table 3 and Sec. 4.5. Scaling up the approach with much more data to train a multi-task generalist policy is a promising direction for future work.
> - **35 demonstration count**: Note that the evaluation episodes have randomized active-object and distractor-object poses, so there is significant variation that the policy needs to generalize over. The use of 35 demonstrations for training does not indicate that the evaluation evidence is limited. Rather, it actually shows our method is sample-efficient enough to perform well over the different variations with a relatively small amount of training data.
> - **20 trials sample size**: We note that 20 is a fairly large number of trials for real-world robot evaluation and requires a large amount of time/effort to evaluate over all the different policies and tasks. In comparison, many other works such as General-Flow (Yuan et al. 2024) and ZeroMimic (Shi et al. 2025) evaluate using 10 trials. Overall, the results show that 20 trials is more than enough to provide answers to the research questions we are interested in.
> - **6-DoF pose limitations**: We agree that 6-DoF pose cannot fully model the motion of articulated and deformable objects, and we discuss this in Sec. 5. However, we also note that many real tasks only require manipulating objects that are rigid or approximately rigid, so such capabilities already have high economic value.
> - **ICP failures under occlusion**: We do not intend the pose tracking methods to be part of our contribution. Our framework can be applied with any pose tracking method and should benefit from the future development of stronger pose tracking methods. Furthermore, one of the benefits of our Simulate step is the ability to filter out the tracking failures so that they do not affect training.
>
> We will upload a revised PDF with the described changes shortly. Please feel free to follow up on this thread in the meantime!

---

### Comment · Action_Editor_ZKpC · 2026-04-11
**Discussion Period**

All reviews have been made public. Authors and reviewers are encouraged to engage in a discussion to address any questions or concerns raised in the reviews. This phase will last ~2 weeks, ending on 24th April.

Reviewers: please use this discussion period to gather any additional information you need to feel confident in your decision recommendation. You will be able to submit your formal recommendation in 2 weeks.

Authors: please read the reviews carefully and respond to any questions or points raised by the reviewers.

Best,

AE

---

### Author Response · Authors · 2026-04-20

The paper PDF is now updated with all the changes described in our replies.

---

### Decision · Action_Editor_ZKpC · 2026-05-18

**Recommendation:** Accept as is

**Audience:**

Yes

**Audience Explanation:**

Learning manipulation skills from human videos is a topic of broad interest to the robot learning community, a significant portion of TMLR's readership. The simulation-filtering approach addresses a practical and well-motivated gap.

**Claims And Evidence:**

Yes

**Claims Explanation:**

The core claim, that simulation-filtered, task-oriented grasping outperforms naive grasp generation, is clearly supported by real-world experiments on an xArm7 across multiple tasks, with quantitative comparisons against baselines.